# 3D-Printing to Plan Complex Transcatheter Paravalvular Leaks Closure

**DOI:** 10.3390/jcm11164758

**Published:** 2022-08-15

**Authors:** Vlad Ciobotaru, Victor-Xavier Tadros, Marcos Batistella, Eric Maupas, Romain Gallet, Benoit Decante, Emmanuel Lebret, Benoit Gerardin, Sebastien Hascoet

**Affiliations:** 1Structural and Valvular Unit, Hôpital Privé les Franciscaines, 7 Rue Jean Bouin, 30000 Nîmes, France; 2Hôpital Marie Lannelongue, Groupe Hospitalier Paris Saint Joseph, Faculté de Médecine Paris-Saclay, Université Paris-Saclay, Inserm UMR-S 999, BME Lab, 133 Avenue de la Résistance, 92350 Le Plessis Robinson, France; 3IMT, Mines Telecom Institute, 30319 Ales, France; 4Cardiology Unit, Hôpital Henri Mondor (AP-HP), 51 Avenue du Maréchal de Lattre de Tassigny, 94010 Créteil, France

**Keywords:** paravalvular leak, prosthetic valve, interventional cardiology, percutaneous, transcatheter, 3D printing, multimodality imaging

## Abstract

Background: Percutaneous closure of paravalvular leak (PVL) has emerged as an alternative to surgical management in selected cases. Achieving complete PVL occlusion, while respecting prosthesis function remains challenging. A multimodal imaging analysis of PVL morphology before and during the procedure is mandatory to select an appropriate device. We aim to explore the additional value of 3D printing in predicting device related adverse events including mechanical valve leaflet blockade, risk of device embolization and residual shunting. Methods: From the FFPP registries (NCT05089136 and NCT05117359), we included 11 transcatheter PVL closure procedures from three centers for which 3D printed models were produced. Cardiac CT was used for segmentation for 3D printed models (3D-heartmodeling, Caissargues, France). Technology used a laser to fuse very fine powders (TPU Thermoplastic polyurethane) into a final part-laser sintering technology (SLS) with an adapted elasticity. A simulation on 3D printed model was performed using a set of occluders. Results: PVLs were located around aortic prostheses in six cases, mitral prostheses in four cases and tricuspid ring in one case. The device chosen during the simulation on the 3D printed model matched the one implanted in eight cases. In the three other cases, a similar device type was chosen during the procedures but with a different size. A risk of prosthesis leaflet blockade was identified on 3D printed models in four cases. During the procedure, the occluder was removed before release in one case. In another case the device was successfully repositioned and released. In two patients, leaflet impingement was observed post-operatively and surgical device removal had to be performed. Conclusion: In a case-series of complex transcatheter PVL closure procedures, hands-on simulation testing on 3D printed models proved its usefulness to plan and facilitate these challenging procedures.

## 1. Introduction

Transcatheter closure of paravalvular leaks (PVL) has emerged as an alternative to surgical management in selected cases [1,2,3,4,5]. A self-expanding occluder with a memory of shape property is deployed within the leak. The main challenges are achieving complete leak occlusion, achieving stable occluder implantation while maintaining valve prosthesis function [3]. A wide spectrum of devices is used, of which two types are specifically labelled for this procedure, namely the paravalvular leak occluder (PLD) (Occlutech GmbH, Jena, Germany) and Amplatzer paravalvular leak plug (AVP) 3 (Abbott Medical, Plymouth, MN, USA) with multiple device sizes available [6,7]. A multimodal imaging analysis before the procedure and transesophageal echocardiography during the procedure allows to choose one of these devices based on PVL shape and measurements [8,9]. Device selection is a crucial step for procedural success. Indeed, complications have been reported following these procedures including mechanical valve leaflet blockade, device embolization, mechanical hemolysis when residual shunt occurs and pericardial effusion [4,5,10]. Similarly, hands on simulation on 3D printed models has emerged as a useful tool to plan complex cardiac interventions [11,12,13,14,15,16].

These 3D printed models incorporate all the anatomical variables, the geometry of the defect but also the surrounding features of the valve and offers devices testing. Already, pre-procedural tests on 3D printed models have shown to be useful to plan transcatheter left atrial appendage closure [17], percutaneous pulmonary valve implantation [12] and transcatheter correction of sinus venosus atrial septal defect [18]. We aim in this study to assess the benefit of hands-on simulation on 3D printed models to plan complex transcatheter PVL closure.

## 2. Methods

We retrospectively selected PVLc procedures in which 3D printed models were used prior to the procedure from the prospective FFPP registry that included 238 procedures in 213 patients between 2017 and 2019 (NCT05089136 and NCT05117359). A single operator (Vlad Ciobotaru, 3D-heart modeling, Caissargues, France) performed all 3D modeling and 3D printed models. We collected clinical data, anatomical data, procedural data and outcomes. A descriptive analysis was performed. All patients had a multimodality imaging analysis including 3D transesophageal echocardiography, cardiac CT and a 3D printed model based on the CT scan.

### 2.1. The 3D Printing Process

A multiphase cardiac CT was performed before the procedure which was used for segmentation (Philips IntelliSpace Portal V11, Best, The Netherlands) based on an intelligent recognition of the tissues from point to point, including muscle and wall parts, enhanced cavities and the cardiac prosthesis and surrounding calcifications. A volume rendering was obtained from segmented structures and was then transformed into a stereolithography (STL) file. Post processing of the STL file was performed offline: for smoothing, hollowing, trimming or thickening (Figure 1). We employed powder-based 3D printing technology that uses a laser to fuse very fine powders (TPU, thermoplastic polyurethane) into a final part-laser sintering technology. The elasticity of the 3D model was adapted according to the selecting-laser-sintering process parameters and the thickness of each component.

### 2.2. Simulation and Testing on 3D Printing Models

A set of devices including PLD Rectangular Waist and AVP 3, Amplatzer Septal Occluder (Abbott Medical, Plymouth, MN, USA), Amplatzer Vascular Plug 2 (Abbott Medical, Plymouth, MN, USA), Amplatzer Duct Occluder (Abbott Medical, Plymouth, MN, USA), Muscular VSD Occluder (Abbott Medical, Plymouth, MN, USA) was used to test PVL closure on the 3D printed models. A Sapien valve 29 mm in diameter (Edwards Lifesciences, Irvine, CA, USA) was used to simulate valve-in-valve and valve-in-ring as an alternative strategy. The risk of leaflet impingements, device embolization and residual leak were investigated during the manipulations.

## 3. Results

A total of 11 patients were included from three centers (Table 1). Patients had mitral prostheses in four cases, of which three were mechanical; aortic prosthesis in six cases of which two were bioprosthesis, two were Perceval sutures-less valves, two were mechanical valves. A transcatheter PVL closure was performed post tricuspid valve-in-ring implantation.

**Table 1 jcm-11-04758-t001:** Anatomical characteristics, operative strategies employed and outcome of cases with 3D printing simulation.

Type of Prosthetic Valve	Location of the Paravalv Location of the Paravalvular Leakular Leak	Size of the Defect on 3d stl Model	Unsuitable Device: 3d Printing Model	Suitable Device: 3d Printing Model	Expected Difficulties	Device Implanted	Device Removed	Difficulties Observed	Outcome
Mitral Mechanical (Case 1; Figure 2, Figure 3 and Figure 4)	Posterior septal	5 o’clock: 8 × 3 mm	AVP2 8, ASO 4, ADO2 6-4, AVP3 10	AVP3 12-3	Leaflet blockade	AVP3 12-3	0	Temporary leaflet blockage Solved by repositioning of the device	Success: Minor residual leak
Mitral Mechanical (Case 2; Figure 5)	Antero-septal	2 o’clock: 12 × 5 mm	AVP2, AVP3 10, ASD, PLD 14W	AVP3 14-5	Leaflet blockage and Residual leak	AVP3 10-5 + AVP2 10 + AVP2 12	ASD	Residual leak solved by 2 additional devices	Success: Minor residual leak
PVL Aortic Perceval Valve (Case 3; Figure 6)	Medial invagination of the basal cage	8 o’clock: 10 × 7 mm	AVP2, AVP3	SAPIEN 23	Left Coronary obstruction	SAPIEN 23 valve in valve	0	Recurrence of invagination after non-compliant balloon	Success: No leak
Aortic Mechanical (Case 4; Figure 7)	Complex tunnel: Peri aortic abscess	7 o’clock: 13 × 5 × 4 mm	AVP2, AVP3 10, AVP3 14, VSD	AVP3 12-3	Leaflet blockage and Residual leak	AVP3 14-5	0	Secondary leaflet blockage	Emergency surgery
Mitral Mechanical (Case 5; Figure 8)	Inferior	6 o’clock: 11 × 4 mm	AVP2 9, AVP2 10, AVP2 12	AVP3 14-3	Leaflet blockade	AVP3 14-5 + AVP3 12-5	0	Secondary leaflet blockage	Emergency surgery
PVL Tricuspid Annular ring (Case 6; Figure 9)	Antero-septal	15 × 7 mm	AVP 3, AVP 2	SAPIEN 29 valve in ring and 2x AVP2	Embolization /Residual leak	SAPIEN 29 + AVP2 14 + AVP2 14	0	0	Succes: No leak
Mitral Bioprosthesis	Posterior inferior	6–9 o’clock: 14 × 3 mm	AVP 2	AVP3 14-5 x3	Residual leak	AVP3 14-5 + AVP3 14-5	0	0	Success: Minor residual leak
Aortic Mechanical	Double leak: anterior and antero-lateral	3 o’clock: 12 × 3 mm and 6 o’clock: 9 × 4 mm	AVP 2, AVP 3	VSD 8 AVP3 VSD6	Residual leak	VSD 6, AVP3, VSD 8	0	Residual leak	Success: Minor residual leak
PVL Aortic Perceval Valve	Lateral:invagination of the basal cage	3 o’clock: 7 × 5 mm	AVP2, AVP3	SAPIEN 23	Residual leak	SAPIEN 23 valve in valve	0	0	Success: No leak

**Figure 2 jcm-11-04758-f002:**
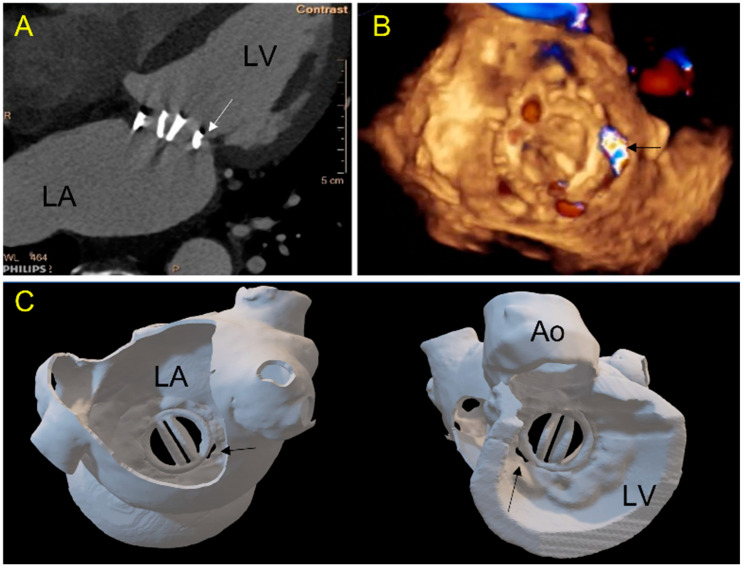
Case 1: A 71-year-old patient with a mechanical mitral valve and a large single postero-septal paravalvular leak (PVL). (**A**) 2D-CT view: a gap (arrow) is seen between the prosthetic ring and the ventricular wall; (**B**) 3D TEE view of the mitral valve from the left atrium showing in 3D color-doppler a posterior septal PVL (arrow); (**C**) 3D printed model showing the double-leaflets mitral mechanical prosthesis in atrial (**left panel**, in a same view as (**B**)) and ventricular view (**panel right**). The PVL location and morphology are well depicted. (LA-left atrium, LV-left ventricle, Ao-aorta).

**Figure 3 jcm-11-04758-f003:**
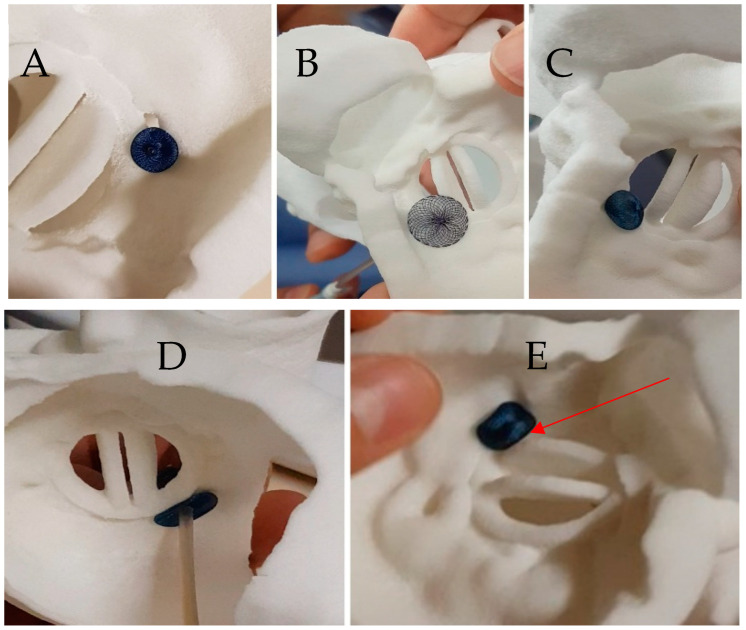
Case 1: Testing of different plugs on the 3D printed model. (**A**) An 8 mm Amplatzer Vascular Plug II with adequate compression but residual leakage is suspected; (**B**) Amplatzer Septal Occluder 4 mm: interference with the mitral prosthesis leaflet (see Appendix A); (**C**) Amplatzer Duct Occluder 2: 6-4, unstable when performing a tug test (**D**) Amplatzer Valvular Plug III 12-3: good apposition of the device without residual gap on atrial view (see Appendix A), (**E**) Amplatzer Valvular Plug III 12-3 in ventricular view showing a close contact with the disc depending on the orientation of the device (see Appendix A).

**Figure 4 jcm-11-04758-f004:**
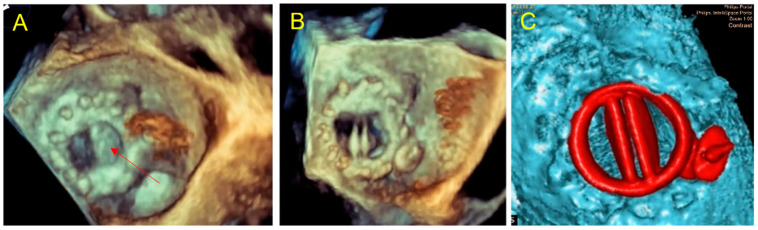
Case 1: per procedural imaging. (**A**) Per procedural 3D TEE in atrial view: Amplatzer Valvular Plug III 3–12 mm blocks the internal prosthetic disc (red arrow); (**B**) traction of the device to adjust its positioning allowing to free the prosthetic disc movements. The position is similar to the simulation on 3D printed model; (**C**) late control CT, good device positioning without leaflet impairment, no residual mitral paravalvular gap, the plug is located as on 3D printed model simulation.

**Figure 5 jcm-11-04758-f005:**
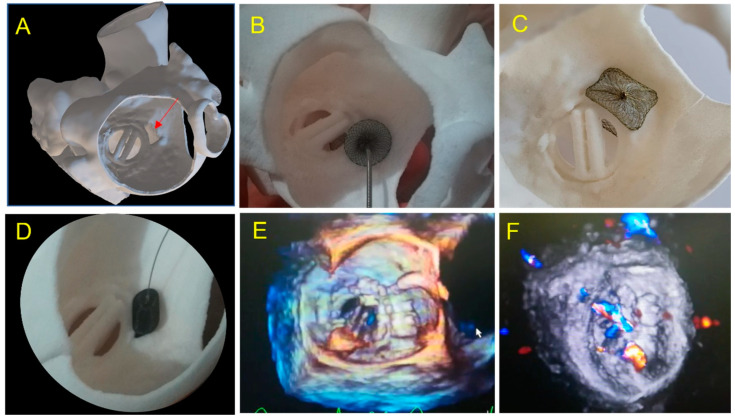
Case 2: A 67-year-old patient with a large paravalvular mechanical mitral leak. (**A**) 3D printed model showing a large PVL (red arrow) in atrial view in an antero-septal position; (**B**) testing of an Amplatzer Septal Occluder device showing prosthetic leaflet impairment; (**C**) testing of a Paravalvular Leak Device Rectangular 14 Waist. Position and defect sealing seem adequate, but risk of disc interference is suspected; (**D**) testing of an Amplatzer Valvular Plug 3 14 × 5 mm. Optimal result is expected; (**E**) 3DTEE view: Amplatzer Valvular Plug III 10 × 5 mm was inserted with a residual leak requiring a second AVPII 10 mm placed subsequently; (**F**) two vascular plug: AVP III and AVPII 10mm with minor inferior residual leak.

**Figure 6 jcm-11-04758-f006:**
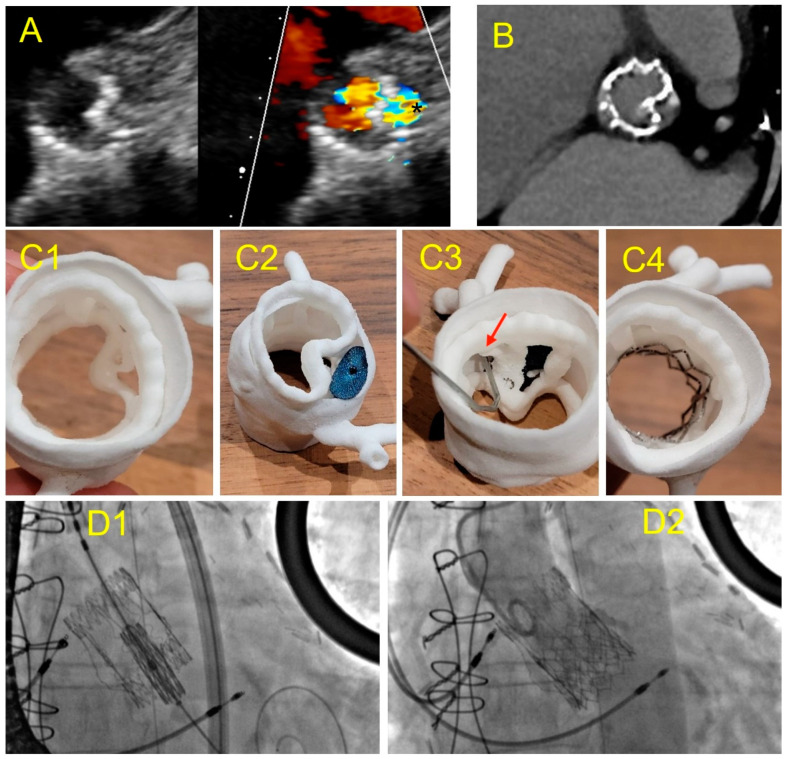
Case 3: PVL after Perceval Bio-prosthesis implantation. (**A**) Aortic TTE view: incomplete expansion of the prosthetic framework against the annular aortic wall (left) causing large PVL (*); (**B**) cardiac CT of the Perceval prosthesis showing an invagination of the prosthetic armature in a same view as (**A**); (**C**) 3D printed model in superior aortic view showing the gap between the aortic wall and the invaginated prosthetic armature (**C1**); testing of an Amplatzer Valvular Plug III 14 × 5 mm (**C2**) showing residual gap and obstruction of the left coronary ostium ((**C3**)-red arrow); simulation of a TAVI valve-in-valve which allowed complete apposition of the Perceval framework to the aortic wall (**C4**); (**D**) TAVI valve-in-valve procedure: implantation of an Edwards balloon expandable prosthesis (**D1**) into Perceval armature without residual leakage (**D2**).

**Figure 7 jcm-11-04758-f007:**
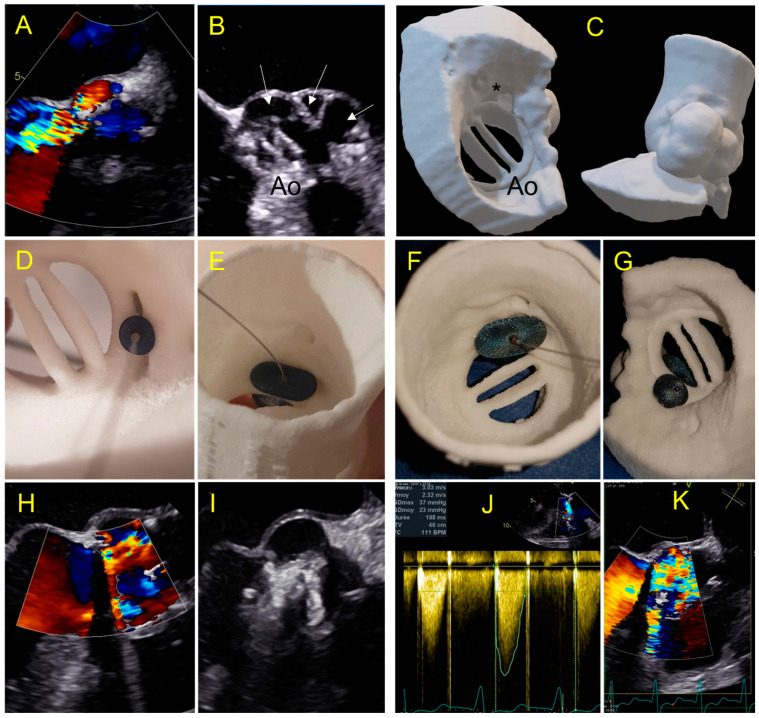
Case 4: complex aortic PVL secondary to paravalvular abscess. (**A**) Aortic PVL in color doppler TEE view; (**B**) multilobed abscess (white arrows) with large PVL.; (**C**) 3D printed model reproducing the multilobed abscesses and the paravalvular leakage with a complex tract with a with a ventricular outlet (*); (**D**,**E**) testing of an AVP 2–10 mm device and AVP 3-10 × 5 mm with a residual PVL (**E**); (**F**,**G**) testing of a 14 mm Amplatzer Valvular Plug 3 device which occludes the leak but has an interference with the leaflet (**G**); (**H**,**I**) procedural implantation of AVP 3 14 × 5 mm with no residual leakage and normal flow through the aortic prosthesis; (**J**,**K**) day one: Increased trans prosthetic gradient with turbulent aliasing flow (**K**) due to a leaflet blockage requiring rapid surgical management.

**Figure 8 jcm-11-04758-f008:**
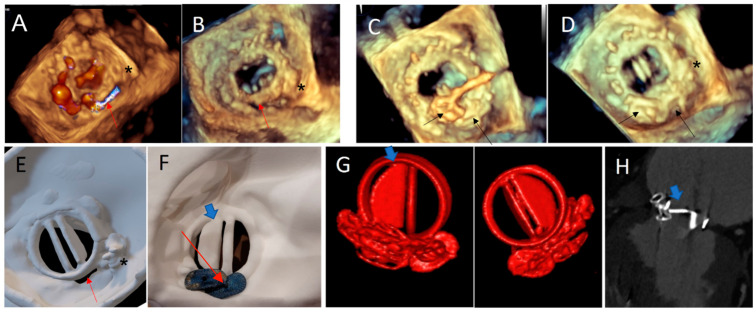
Case 5. (**A**,**B**) Paravalvular residual leak posteriorly (red arrow), in a patient with a previous history of transcatheter PVL closure treated by an AVP3 3–12 mm device (*): 3D TEE views; (**C**) PVL Closure with two additional AVP3 devices: 5–14 and 5–12 mm (black arrow); (**D**) normal opening of the mechanical leaflets and no residual gap at the end of the procedure on 3D TEE view. Note the two AVP 3 devices (black arrow) and the previous one (*); (**E**) atrial view of the 3D printed model showing the PVL close to the hinge of the mechanical mitral discs. Note the 3D print of the pre-existing AVP 3 (*); (**F**) post procedural simulation on 3D printing model: using the same AVP 3 devices: demonstrating a disc interference (blue arrow) in ventricular view; (**G**) 3D volume rendering of the prosthesis with a partial blockage of the external disc (blue arrow), in both ventricular and atrial view, occurred late in the postoperative period, similar to testing on the 3D printed model. (**H**) Internal Disc blockage in 2D CT view (blue arrow).

**Figure 9 jcm-11-04758-f009:**
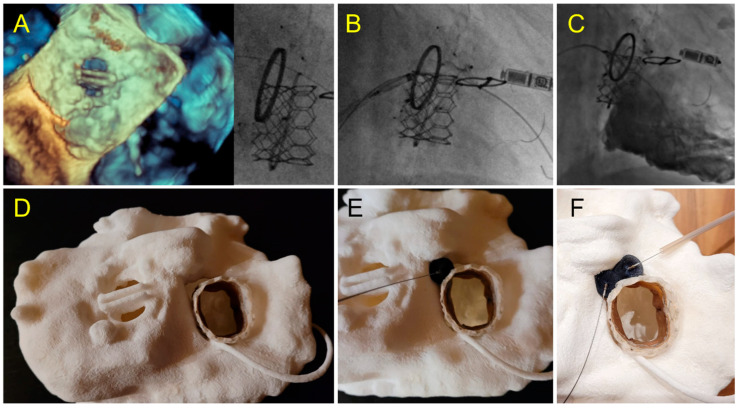
Case 6: Tricuspid PVL. (**A**) Prior Mechanical mitral prosthesis and tricuspid annuloplasty with massive residual tricuspid regurgitation; (**B**) valve-in-ring procedure with Sapien 3 29 mm implanted in the Carpentier Tricuspid ring 36 mm with a septal superior residual gap closed by the insertion of two AVP2 devices.; (**C**) fluoroscopic view of valve-in-ring prosthesis and two AVP2 14 mm devices without residual tricuspid regurgitation during contrast injection; (**D**–**F**) pre-operative step by step simulation. (**D**) Valve-in-ring implantation. (**E**) A first AVP2 is positioned with residual PVL. Implantation of a second AVP2 with complete closure.

### 3.1. Choice of Size and Type of PVLc Plug

One device was used in six patients, two devices in two patients and three devices in three cases. A device implanted was removed before release in one patient. An AVP 3 plug was implanted in 6/11 patients and an AVPII was used in two cases in combination with another plug in one case. A VSD plug was used in two patients. The choice of the optimal device used during the simulation on the 3D model matched with the type and size of plug used during the procedure in eight cases (see Table 1 case 1). In one patient, the inserted device was smaller than the one chosen during simulation and this patient required two additional plugs to cover a residual leak (see Table 1, case 2). In three patients, the inserted device was larger than the one chosen during the simulation on the 3D model; two of these patients required surgery for leaflet blockage. (see Table 1 case 4,5).

An alternative strategy to PVL closure with occluder was considered in two cases and successfully applied. A valve-in-valve in sutures-less Perceval prothesis was performed (see Table 1 case 3). In one patient with valve-in-tricuspid ring, two VP2 were implanted (see Table 1 case 6).

### 3.2. Detection of Prosthetic Leaflet Blockage

The prospective 3D printing simulation detected a risk of prosthetic blockage in four cases. Interestingly, in a patient with mechanical mitral PVL, the appropriate choice of the device was an AVP3, but a risk of leaflet blockade was identified that depended on the orientation of this oval device. This issue occurred during the procedure and was solved by proper orientation of the AVP3 in a similar position as during the simulation (Figure 2, Figure 3 and Figure 4). In one case with a large mitral PVL (Figure 5), an ASO that was initially deployed, was removed due to leaflet blockade. In two other patients, the plug-induced blockage was anticipated during the simulation on the 3D model and occurred a few hours after the end of the procedure (Figure 7 and Figure 8) and required emergent surgery for leaflet blockade.

### 3.3. Detection of Residual Shunt

A risk of residual shunt was anticipated in six cases and observed after the procedure in three cases.

The targeted PVL was successfully sealed in 9 of 11 cases with minor residual leak in 4 cases. In one case, the residual PVL was very small and was not expected on the simulation (Figure 3 and Figure 4).

## 4. Discussion

In this multi-center case-series, we illustrate how hands-on simulation testing on 3D printed models can be integrated into the planning process of transcatheter PVL closure in addition to multimodal imaging improving communication within the heart team: Imager/Interventional/Surgeon for feasibility of percutaneous addressing. Although complex to achieve, a 3D printed model of PVL was demonstrated to be feasible and useful to predict device-related adverse events such as residual leak and valve leaflet blockade.

A CT scan is not routinely performed to assess PVL given its limits inherent to valve artifacts. However, integrating these limits, we successfully obtained cardiac segmentation of the valve, the PVL and the surrounding structures with morphological assessment complementary to the one provided by echocardiography. Accurate 3D printing models require adapted cardiac CT protocol to optimize the temporal and spatial resolution. Technical specifications included low-pitch spiral acquisition and thin sections with low incremental steps, paying particular attention to the suppression of dynamic, hardening, or metallic artifacts (beam-hardening artifact) by adapting windowing, adjustments of specific algorithms and beam energy, but also by performing additional hard-filter reconstructions and iterative post-processing reconstructions [19]. The process of segmenting the heart is time consuming and may be a limitation of the technique if not performed in conjunction and agreement with other imaging techniques especially 3D TEE. An accurate, fully automated, CT anatomical segmentation would facilitate the process and reproducibility [20].

The appropriate choice of materials and printing techniques are another important point. Multi-material 3D printed models have already been used to explore the role of annular calcification in PVL formation post-TAVR in a retrospective study by Zen Qian et al. by using 3D printed tissue-mimicking phantoms of aortic root, in vitro [21].

SLS technique has been chosen because of its very high spatial resolution and efficient cost–quality ratio [22]. The SLS technique has the advantage of being executed without separate feeder for support material. This allows the printing of complex geometries that were not feasible with other techniques. Moreover, the thickness can be as thin as 0.5 mm, allowing a high flexibility for the aortic wall, whereas a larger thickness makes the material stiffer to mimic a prosthetic ring or leaflet. The TPU material is tear-resistant, permitting leaflets movement around the hinge points.

Three-dimensional printing has the advantage of displaying all the elements required to adequately select an occluder that will match with the complex PVL shape. Location, size, number and morphology of PVL, together with the relationships with the valve and the surrounding structures, are depicted in a condensed manner in the 3D printed models [15,16,22,23].

The PVL closure procedures correspond to very diverse, sometimes complex situations, with their own specificities depending on the type of prosthesis and the PVL location. The strategy and pre-operative planning are of paramount importance with a tailored approach facilitated by 3D model testing.

In our series, the prediction of the risk of leaflet blockade was anticipated in all cases during testing, whereas it was uncertain using imaging alone. Indeed, interference with the mitral prosthesis can occur on the ventricular side, which is not accessible to direct analysis by TEE. Furthermore, the expansion of the plug and its alignment with the axis of the prosthetic wing is hard to predict solely by imaging., depending strongly on the shape of the PVL.

Interestingly, in two patients, the obstruction occurred a few hours after the procedure despite a result appearing optimal with a normal symmetrical movement of the leaflet initially at the end of the procedure, requiring a secondary surgical valve replacement. This delayed leaflet blockage was due to a change in preload conditions and not because of a displacement or migration of the prosthesis which was confirmed during the per-surgical examination showing an adequate localization of the PVLc device. It should be noted that all PVLc procedures were validated by a heart team and planned on TEE and preoperative CT, however this obstruction risk had been anticipated solely by the 3D printing simulation and dynamic testing using the same device but not by imaging alone.

Very complex, unusual hybrid strategies have been planned on 3D models. A severe tricuspid regurgitation was treated by SAPIEN valve-in-ring implantation. The ring was semi-circular and a residual severe PVL was expected on the simulation, adequately resolved by implantation of two VP 2 in a strategy similar to what was planned on pre-procedural testing. In this case, simulation on 3D printing gives us confidence of the feasibility of this complex strategy and makes the procedure fast and straightforward following the plan. Two other cases of PVL on Perceval valves due to stent infoldings, illustrate the importance of preoperative testing for procedure planning. In these cases, the simulation on the 3D printed models allowed us to identify that the valve-in-valve strategy would adequately address the issue versus a plug implantation, which was shown as unstable, inefficient and with risk of obstruction towards the coronary ostium. The simulation allowed a fast and efficient procedure without losing time, energy and devices in a wrong approach.

Additional costs per model are required, however, these costs could be balanced by the reduction in the number of devices used and the reduction in adverse events in complex cases [23]. When considering the cost of a 3D printed model, a global cost-analysis of the procedure should be performed. On-screen simulation is being developed for many cardiac interventions. However, simulation on 3D printed models, in addition to providing a unique comprehensive anatomical analysis, allows to understand the interaction between devices, the valvular prosthesis and adjacent structures taking into account the deformations of the devices and the models that cannot be assessed by the usual imaging modalities [24].

## 5. Limits

We present a limited series of cases. However, to our knowledge, this is the largest study to date to investigate the additive value of 3D printing to plan PVL closure. Furthermore, the diversity of cases illustrates the usefulness of 3D printing planning in a wide spectrum of PVL closure.

The simulation on the 3D model was not standardized, given continuous improvement in materials and techniques for 3D printing as well as evolution in PVL closure techniques.

The segmentation of the CT and 3D printing models were performed by a single operator cumulating experience in cardiac imaging, 3D printing and interventional cardiac echocardiography. Reproducibility of the technique with other operators remains to be demonstrated.

Larger, prospective, multicenter studies should confirm the contribution of 3D printing to multimodality imaging in complex PVL closure procedures.

## 6. Conclusions

In a case-series of complex transcatheter PVL closure procedures, hands-on simulation testing on 3D printed models proved its usefulness to plan and facilitate these challenging procedures.

## 7. Key Question

Prosthetic paravalvular leaks leading to heart failure and/or hemolysis can be treated by interventional cardiology or open-heart surgery. Transcatheter closure remains challenging, and 3D printed models may be helpful in selected cases to guide these complex procedures.

## 8. Key Findings

Hands-on simulation testing on 3D printed models predicted the risk of device related adverse events such as leaflet blockage or residual paravalvular leak.

## 9. Take-Home Message

Three-dimensional printing can be integrated into the multimodal planning of transcatheter paravalvular leaks closure procedures to improve outcomes.

## 10. One-Sentence Summary

A series of 11 transcatheter paravalvular leaks closure cases were planned on 3D laser sintering printed soft TPU model, permitting device implantation testing and prediction of leaflet blockage or residual leak.

## Figures and Tables

**Figure 1 jcm-11-04758-f001:**
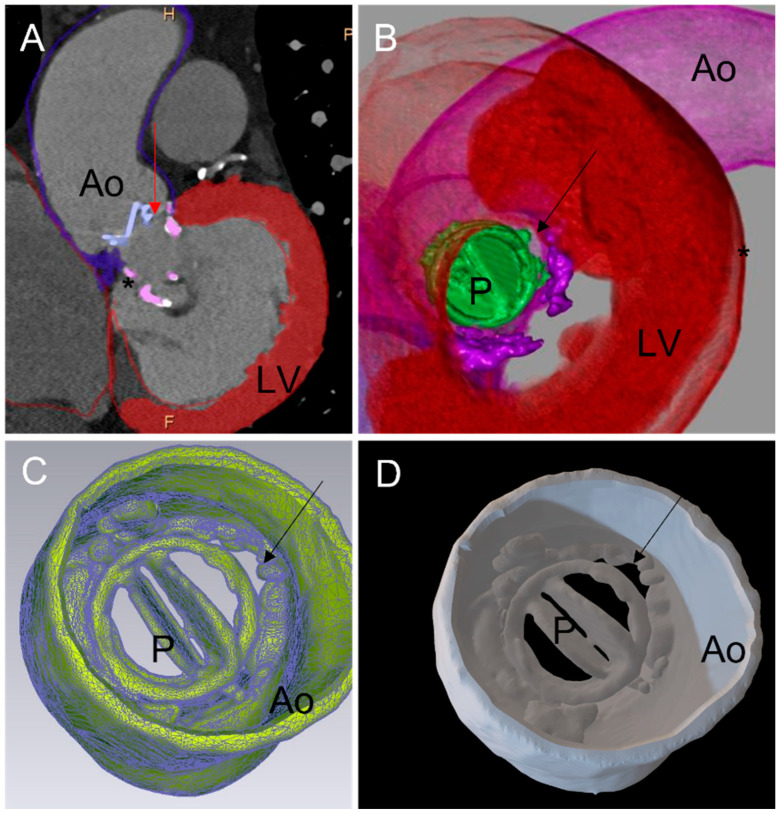
Three-dimensional printing workflow: (**A**) 2D cardiac tomography (CT). Each cardiac structure of interest has been segmented: aortic wall, mechanical aortic valve prothesis, left myocardium, annular calcification (*). A diastasis is observed between the mechanical aortic prosthesis (P) and the aortic wall (Ao) corresponding to a paravalvular leak (PVL) (red arrow). (**B**) CT full volume rendering of the segmented structures. The mechanical aortic valve is displayed in green. The PVL is seen from a left ventricular view. (**C**) A standard triangle language (STL) file was created from segmented structures. The PVL is seen from the aorta (black arrow). (**D**) Three-dimensionally printed model derived from the STL. Visualization of the aortic root, aortic valve (P) and the PVL (black arrow). PVL: paravalvular leak; Ao: aortic; LV: left ventricle; P: prosthesis.

## Data Availability

Data supporting reported results can be found at Hôpital Marie Lannelongue.

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
