# Peer review of "3D-Printing to Plan Complex Transcatheter Paravalvular Leaks Closure"

_jcm, 2022, doi:10.3390/jcm11164758_

Round 1

Reviewer 1 Report

Congratulations for putting all the cases together and the nice reiviw

I miss some complementary material in video format to show how the leaflet block occurs on the 3d model with the occluders

Author Response

We thank the reviewer for his comment

Additional videos have been added as supplementary data

Reviewer 2 Report

Thank you for the opportunity to review the manuscript. Ciobotaru et al have used novel 3D printing technology based on pre- operative CT imaging, for defining the exact location, anatomical relationship, potential interference with the valve leaflets and choosing an appropriate device for correcting paravalvular leak. 3-D printing has been gaining traction in a lot of complex cardiac and interventional procedures. Using this technology for paravalvular leak patients  planning makes perfect sense as they usually have complex and varied anatomy. Just few comments

- In two patients who needed emergency surgery, knowing that they were high risk for leaflet interaction, should they have been referred for surgery in first place or there was  device migration causing the leaflet blockage?

- comparison with traditional patients who did not utilize this technology with regards ton number of devices used, residual leaks or other complications would help to decide about the cost effectiveness of this technology

 want to congratulate the authors on a nicely written manuscript. 

Author Response

Author's Reply to the Review Report2

2.1 Review commen

- In two patients ho needed emergency surgery, knowing that they were high risk for leaflet interaction, should they have been referred for surgery in first place or there was device migration causing the leaflet blockage?

Author's Reply: We thank the reviewer for his comments

Both patients who needed emergency surgery, were redux surgery with a high operative risk. PVLc procedure was planned based on TEE and preoperative CT data and validated by a heart team. The result appeared optimal with a normal symmetrical movement of the leaflet on TEE at the end of the procedure. However, both patients required surgery as a result of a delayed leaflet blockage due to a change in preload conditions rather than to a displacement or migration of the prosthesis. During the surgery: the PVLc device was found to be in normal position.

This obstruction risk had been anticipated solely by 3D printing simulation using the same device and dynamic testing but not by imaging alone. This clearly raises the issue of considering the dynamic risk of prosthetic obstruction in the patient management strategy, in addition to the imaging analysis.

This was added in the main text

Both patients were redux surgery with a high operative risk. PVLc procedure was planned based on TEE and preoperative CT data and validated by a heart team. The result appeared optimal with a normal symmetrical movement of the leaflet on TEE at the end of the procedure. However, both patients required surgery as a result of a delayed leaflet blockage due to a change in preload conditions rather than to a displacement or migration of the prosthesis. During the surgery: the PVLc device was found to be in normal position.

This obstruction risk had been anticipated solely by 3D printing simulation using the same device and dynamic testing but not by imaging alone. (inserted 255)

2.2 Review comment

- comparison with traditional patients who did not utilize this technology with regards ton number of devices used, residual leaks or other complications would help to decide about the cost effectiveness of this technology

Author's Reply: We thank the reviewer for his comments

We believe that printing can be cost effectiveness by reducing the number of prostheses used and by avoiding further complications.

However, our series presented is too limited in number and the decision based on the 3d print model was left to the operator's choice. Thus, we can only make assumptions. Also, a multicentre study with a wider use of the technique in a standardized manner including two comparative groups could provide an answer.

Reviewer 3 Report

I congratulate the authors who have presented this innovative report which will arouse interest in readers.

I would like to know if the authors have considered the possibility of biomodeling using finite element analysis to evaluate the impact of calcifications on PVL formation.

This aspect should be included as a future perspective

Author Response

Author's Reply: We thank the reviewer for his comments

The annular calcifications have a central role in the formation of PVL.

3D Modeling using both soft and rigid materials could be an interesting future perspective regarding the PVL formation. In a retrospective study including 18 patients who underwent TAVR, Zen Qian et al , found a high level of accuracy in predicting post-TAVR PVL, in terms of its occurrence, severity, and location by using 3D printed tissue-mimicking phantoms TAVR aortic root, in vitro.  ( JACC Cardiovasc Imaging. 2017 Jul;10(7):719-731. doi: 10.1016/j.jcmg.2017.04.005.)

This was added in the main text